# HIV prevalence, sexual risk behaviours and HIV testing among female sex workers in three cities in Sri Lanka: Findings from respondent-driven sampling surveys

Ariyaratne Manathunge[1], Jelena Barbaric[2], Tomislav Mestrovic[3], Sriyakanthi Beneragama[1], Ivana Bozicevic[2]*

1 National STD/AIDS Control Programme, Ministry of Health, Colombo, Sri Lanka, 2 World Health Organization Collaborating Centre for HIV Strategic Information, University of Zagreb School of Medicine, Zagreb, Croatia, 3 University Centre Varazdin, University North, Varazdin, Croatia

* Ivana.Bozicevic23@gmail.com

## Abstract

Sri Lanka has a low-level HIV epidemic. This study aims to provide evidence on HIV, syphilis and hepatitis B (HBV) prevalence, sexual risk behaviours and utilisation of HIV prevention interventions among female sex workers (FSW) in the cities Colombo, Galle, and Kandy. Using respondent-driven sampling (RDS), we recruited a total of 458 FSW in Colombo, 360 in Galle and 362 in Kandy from November 2017 to March 2018. Participants provided biological specimens for testing for infections and completed a behavioural questionnaire. We found no HIV nor HBV infections in Galle and Kandy, and low HIV (0.4%) and HBV surface antigen (0.6%) prevalence in Colombo. FSW in Colombo had higher positivity on *Treponema pallidum*-particle agglutination test (8.4%) compared to Galle (2.0%) and Kandy (2.5%). About two thirds of FSW heard of HIV in each of the cities. Around 90% of FSW used condom at last sex with a client in both Colombo and Galle, but considerably less in Kandy (57.1%). However, lower proportion of FSW used condoms every time during sex with clients in the past 30 day: 22.9% of FSW in Colombo, 26.6% in Kandy and 68.4% in Galle. Across cities, 17.5%-39.5% of FSW reported being tested for HIV in the past 12 months or knowing HIV positive status. The commonest reasons for never testing for HIV was not knowing where to test (54.2% in Colombo, 41.8% in Galle, 48.1% in Kandy) followed by inconvenient testing location (23.7% in Colombo and 31.1% in Kandy). HIV has not yet been firmly established among FSW in three cities in Sri Lanka, but the vulnerability towards HIV and STIs is substantial. HIV interventions should be intensified by expanding community-based HIV testing approaches, increasingawareness of HIV risks and addressing socio-structural vulnerabilities of FSW to HIV.

**Data Availability Statement:** All relevant data are within the manuscript and its Supporting Information files.

**Funding:** The study was funded by the New Funding Model Grant, 2016-2018 of the Global Fund to Fight AIDS, Tuberculosis and Malaria (https://www.theglobalfund.org/en/). The funders had no role in study design, data collection and analysis, decision to publish, or preparation of the manuscript.

**Competing interests:** The authors have declared that no competing interests exist.

## Introduction

Sri Lanka has a low-level HIV epidemic, with HIV prevalence estimates below 0.1%, which is lower than in most South Asian countries [1]. In 1930's it was one the first in Asia Pacific to introduce the universal health coverage and its health system has a long track record of strong performance, despite 30 years of armed conflict which ended in 2009 [2]. This multi-ethnic, middle income country has a population of 21.7 million [3]. An estimated 3500 (3100–4000) people were living with HIV in Sri Lanka at the end of 2018 [1]. In 2018, 350 HIV infected persons were newly reported nationwide, the highest number in a year since 1987 when the first case was identified [4].

Using mapping and the multiplier methods in 2018, it was estimated that the population size of female sex workers (FSW) is 30,000 (plausibility range 20,000–35,000) [5]. Sex work in Sri Lanka is illegal and highly stigmatised [6]. The results of the first round of an integrated bio-behavioural survey (IBBS) among FSW in 2014/2015 included prevalence estimates for HIV, active syphilis and *Treponema pallidum*-particle agglutination assay (TPPA) positivity of 0.1%, 1.6% and 4.8% in Colombo, respectively [7]. In Galle, HIV prevalence was estimated at 1.0% whereas none of the participants tested positive for syphilis [7]. In Kandy, no HIV or syphilis infections were found [7]. Only about a third of the participants were able to correctly identify modes of transmission of HIV and reject major misconceptions about HIV transmission across the three cities. Whilst estimated 92% FSW used condoms at last sex with a client, only about a third tested for HIV 12 months before the survey and knew their HIV test result [7].

In Sri Lanka, HIV testing in key populations is carried out in sexually transmitted infection (STI) clinics, as well as through outreach testing services [4]. Whilst the number of STI clinic-based HIV testing among FSW rose moderately from 1135 tests performed in 2015 to 1715 in 2018, outreach testing among FSW rose markedly, from 490 tests in 2015 to 4064 tests in 2018 [4].

Being tested and knowing one's HIV status is essential for initiation of ART and achieving the viral load suppression, which leads to improved outcomes for the individual and prevents HIV transmission at the population level [8]. FSW continue to bear a higher risk of HIV in many countries and are an important target population for a public health response to HIV [1, 8]. In order to monitor and further characterise HIV-related risk behaviours, uptake of HIV prevention services and HIV prevalence, we performed the second round of IBBS among key populations at risk of HIV. This paper aims to describe the key results of the second IBBS among FSW in Colombo, Galle and Kandy, assess correlates of not being tested for HIV and compare the key findings with those from the first IBBS carried out in 2015. Colombo, Galle and Kandy were selected for IBBS in 2018 because IBBS were done in these three cities in 2015. We hypothesised that there will be a significant increase in condom use with non-paying partners and ever testing for HIV when 2015 data are compared with 2018. They were selected for IBBS in 2015 because they are in the most populous provinces in Sri Lanka named Western Province (Colombo), Central Province (Kandy), and Southern Province (Galle) and because of the estimated population size of FSW. By repeating surveys in the same cities over time, temporal changes in HIV indicators can be assessed.

According to the findings of the latest Demographic and Health Survey in Sri Lanka done in 2016, which included ever married women aged 15–49, there are some substantial socioeconomic differences among these three cities [9]. The percentage of women that cannot read and write was 3.4% in Colombo, 4.9% in Galle and 8.1% in Kandy and employed at the time of DHS were 39.0%, 38.2% and 31.5%, respectively. Experience of any form of domestic violence on a daily basis was reported by 16.3% of women in Kandy followed by 10.8% in Galle and 6.0% in Colombo.

## Materials and methods

### Study design, setting and participants

As FSW in Sri Lanka are a hard to reach population, we used respondent-driven sampling (RDS), a peer referral method which is widely used to sample such populations at risk for HIV [10–13]. Furthermore, RDS was also used in the last IBBS survey of FSW in 2015, which allows for data comparison [7]. The results of the formative assessment, which included key informant interviews and focus group discussions in cities where RDS survey was planned, had revealed that the RDS would be feasible. We obtained ethical approval for the study from the Medical University of Sri Jayewardenepura in October 2017. Recruitment and data collection lasted from November 2017 to March 2018. Surveys took place at premises of non-governmental organisations (NGOs) that are providers of HIV prevention services to FSW.

Eligibility criteria included a female who has sold sex in exchange of money or goods in the six months before the survey, was older than 18 years of age, was able to provide verbal informed consent and resided or worked in the city where IBBS is done for at least 12 months before the survey. Personal network size question was worded as follows: "How many women do you know (they know your name and you know their name), who have sold sex in the last 6 months, and who are older than 18 years and reside in this city?" We identified initial respondents (seeds) during the formative assessment and started recruitment in Colombo with four, in Galle with two and in Kandy with seven seeds. Seeds were instructed to recruit with coupons given by study staff up to three other FSW from their social circle, who in turn were enrolled (if eligible) and instructed to refer other peers with coupons. Each coupon had its unique number that enabled to link participants with those who gave them coupons to participate in the study. Coupon numbers enable to monitor recruitment chains and issue secondary incentives, and are also necessary for RDS data analysis. Coupons also contained information about the address of the study site and time-period when the study site is open.

Participants provided informed consent before participating and were screened for eligibility.

After they gave an oral informed consent to participate, a survey staff member signed an informed consent form on their behalf. The Ethics Committee approved this consent procedure.

After completing a questionnaire, participants received HIV pre-test counselling and afterwards provided biological specimens for testing for infections. All participant information was kept anonymous and confidential.

Participants received a primary incentive of 2 USD for their participation in the study, and a secondary incentive of 1 USD for each person they recruited and who successfully completed the survey. There were no changes in the number of coupons per recruiter or any other relevant variation in study procedures during data collection.

Sample size calculation was set up to detect a 15% increase in consistent condom with a non-paying partner during 30 days before the survey from the baseline level of 32% in Colombo, 34% in Galle and 50% in Kandy in 2015, with 80% power and an alpha error of 5%. This gave a sample size of 442 respondents in Colombo, 307 in Galle and 341 in Kandy, which was rounded to 450 FSW in Colombo, and 350 FSW in Galle and Kandy, respectively.

### Survey data collection and laboratory testing

We used behavioural questionnaire designed for the IBBS in 2014 and modified it according to the WHO-recommended questionnaires for bio-behavioural HIV surveys in key populations [14]. Indicators needed for the Global AIDS Monitoring (GAM) were collected based on

the questionnaire [15]. The questionnaire included questions on demographics, sexual and drug-injecting behaviours, HIV testing, HIV-related knowledge, uptake of HIV prevention services and experience of violence. Indicators on sexual behaviours included age of the first vaginal and anal sex, length of time working as a sex worker, condom use at last sex with clients and regular partners, and consistent condom use with clients and regular partners during 30 days before the survey. Coverage with HIV interventions was defined according to the GAM guidelines, as receipt of at least two out of the following HIV prevention services from an NGO or a health care provider in the past three months: condoms and lubricants; counselling on condom use and safe sex; testing for sexually transmitted infections (STIs). The questionnaire was administered in a face-to-face interview using a tablet with Open Data Kit. The recruiter-recruit relationship was tracked using unique coupon numbers, which were entered into an MS Excel-based database.

We conducted testing for HIV, syphilis and hepatitis B virus (HBV) using rapid, point-of care tests on site from an intravenous blood draw. We performed serial HIV testing in line with the WHO and the national algorithm for HIV testing, using Alere Determine HIV (Alere Determine HIV-1/2 Ag/Ab Combo, Abbott Laboratories, USA) rapid test kit as the first, and in case of reactivity SD Bioline HIV 1/2 rapid test (SD Bioline HIV-1/2 3.0, Global Diagnostics, India) as the second, and ABON Tri-Line HIV Rapid Test (ABON Biopharm Hangzhou Co., China) as the third test. Determine Syphilis TP rapid test (Alere Determine™ Syphilis TP, Abbott Laboratories, USA) was used as the first test for syphilis, and those with reactive test results were re-tested using Venereal Disease Research Laboratory (VDRL) test (titre >8 was considered indicative of active syphilis). For hepatitis B testing, we used Determine HBsAg rapid test (Abbott Laboratories, USA) to determine the presence of the HBV surface antigen (HBsAg). Confirmatory testing was performed at the National Reference Laboratory of the National AIDS/STD Control Programme (NSACP). For quality assurance purposes, every 10th HIV and syphilis-negative sample was re-tested at the National Reference Laboratory of the NSACP. All respondents with confirmed positive test results were referred to the nearest STI clinic for further clinical assessment.

## Statistical methods

We calculated the sample and estimated population proportions with 95% confidence intervals using RDS Analyst software version 0.61 [16]. We used the Gile's sequential sampler (SS) estimator that is based on the inclusion probabilities that are calculated using participants self-reported network sizes, recruitment patterns and estimated population size [17]. According to the population size estimation data collected in 2013, Gile's SS with population size estimate of 6,157 was used for Colombo, 1,754 for Galle and 2,204 for Kandy along with 0.95 confidence intervals and 5,000 bootstraps.

We analysed homophily, convergence/equilibrium and bottlenecks for key indicators (knowledge of HIV status, HIV prevention coverage, condom use at last commercial sex, discriminatory attitudes towards PLHIV, avoidance of stigma and discrimination, age and income [18].

We used a Chi-square significance test to compare selected categorical HIV indicators collected in the IBBS carried out in 2018 with those collected in the 2015 IBBS.

We used Stata 16.1 to perform unweighted bivariate and multivariate analysis [19]. Unweighted regression analysis was used due to concerns that weighted regression of RDS data does not perform well [20]. Multivariate logistic regression analysis was done to determine correlates associated with not being tested for HIV in the past 12 months or not knowing HIV positive status, which is the GAM indicator for HIV testing among key populations [15].

Independent variables used in the model included socio-demographic factors, sexual behaviours and knowledge about HIV. Knowledge about HIV was categorised as comprehensive knowledge, which meant correct answers on all five questions about HIV transmission outlined in the GAM 2020 Guidance, and non-comprehensive, which meant at least one incorrect answer to the five questions. We included factors associated with the outcome at $p < 0.05$ in the bivariable analysis, as well as age, in the final logistic regression model. We considered a result to be statistically significant in the multivariable analysis at $p = 0.05$. We excluded participants with missing values on any variable in the model. We included initial respondents (seeds) in the analysis.

## Results

### Recruitment patterns

We enrolled a total of 458 FSW in Colombo, 360 in Galle and 362 in Kandy. The predetermined sample size was reached at all three study-sites. The maximal chain length ranged from five in Kandy to nine in Colombo. Convergence was reached on all seven key indicators in Galle and in Kandy, and on five out of seven indicators in Colombo. The homophily on key variables ranged from 0.72 to 1.14 in Colombo, from 0.98 to 1.23 in Galle and from 0.99 to 1.12 in Kandy. The median network size was 10 (range: 1–300) in Colombo, 8 (range: 1–35) in Galle and 7 (range: 1–80) in Kandy.

### Socio-demographic characteristics

The median age of FSW was 40 years in Colombo, 37 in Galle and 41 in Kandy. Median age of FSW recruited in IBBS in 2015 was 37 years in Colombo and Galle and 33 years in Kandy. Never attending school was reported by a minority in Colombo and Galle (6.6% and 5.0%, respectively), but by 19.1% FSW in Kandy. The majority of FSW had only some form of primary school education completed (47.9% in Colombo to 62.9% in Galle).

Earning less than 150 USD monthly was reported by 24.3% of FSW in Colombo, 30.1% in Galle and 45.9% in Kandy. This level of income was in the 2015 IBBS reported by 39.2% of respondents in Colombo, 35.4% in Galle and 17.3% in Kandy. Over 50% of FSW had sources of income other than sex work (Table 1).

### Knowledge, risk-related behaviors and experience

Around two thirds of FSW have heard of HIV in each of three cities, but FSW knowledge about HIV prevention varied, as only 15.0% of FSW correctly answered all five questions about HIV transmission in Colombo, 27.8% in Kandy and 41.5% in Galle (Table 2). Compared to 2015, there has been a decline in comprehensive knowledge in Colombo and Kandy in 2018, and an increase in Galle. Less than 20% of FSW across the cities perceived their personal HIV risk as high.

Median reported age at first vaginal sex was 18 years (range 12–35) in Colombo, 18 years (range 12–40) in Galle and 21 years (range 12–38) in Kandy. Median reported age at first anal sex was 21 years (range 14–48) in Colombo, 19 years (range 14–45) in Galle and 24 years (range 15–48) in Kandy. Median reported length of time working as a FSW was 19 years (range 0–49) in Colombo, 9 years (range 0–46) in Galle and 8 years (range 1–48) in Kandy.

Median total number of paying partners in the past seven days was 6 (range 1–28) in Colombo, 4 (range 0–20) in Galle and 3 (range 1–15) in Kandy.

Around 90% of FSW used condom at last sex with a client in both Colombo and Galle, but considerably less in Kandy (57.1%). However, condoms were reportedly used every time

**Table 1. Sociodemographic characteristics of female sex workers in Colombo, Galle and Kandy, Sri Lanka, 2018.**

| | Colombo | | Galle | | Kandy | |
|---|---|---|---|---|---|---|
| | n/N | % (95% CI)[†] | n/N | % (95% CI) [†] | n/N | % (95% CI)[†] |
| Age groups (in years) | | | | | | |
| 18–24 | 36/457 | 6.8 (4.3, 9.3) | 20/360 | 5.3 (3.1, 7.6) | 27/362 | 8.0 (4.8, 11.2) |
| 25–34 | 100/457 | 20.2 (15.7, 24.9) | 121/360 | 30.3 (25.3, 35.3) | 84/362 | 22.8 (18.4, 27.4) |
| 35–44 | 136/457 | 31.6 (26.0, 37.0) | 117/360 | 31.4 (26.4, 36.4) | 117/362 | 33.4 (28.6, 38.2) |
| ≥ 45 | 185/457 | 41.4 (35.6, 47.3) | 102/360 | 32.9 (27.3, 38.5) | 134/362 | 35.8 (30.2, 41.4) |
| Earns money doing anything other than sex work (has other sources of income) | 244/458 | 56.9 (50.8, 63.1) | 222/357 | 64.7 (59.5, 69.8) | 250/361 | 68.7 (63.7, 73.6) |
| Marital Status | | | | | | |
| Single (Never married) | 58/458 | 15.7 (11.3, 20.0) | 31/359 | 7.7 (5.3, 10.1) | 33/360 | 8.5 (5.3, 11.7) |
| Married | 202/458 | 39.0 (33.4, 44.7) | 259/359 | 77.6 (73.4, 81.6) | 176/360 | 46.9 (41.7, 52.2) |
| Divorced/Separated | 151/458 | 35.1 (29.1, 41.2) | 43/359 | 8.9 (6.3, 11.5) | 88/360 | 25.3 (20.7, 29.9) |
| Widowed | 47/458 | 10.2 (6.8, 13.6) | 26/359 | 5.9 (3.8, 8.0) | 63/360 | 19.2 (14.8, 23.7) |
| Ever been pregnant | 295/457 | 57.7 (51.1, 64.2) | 279/360 | 74.3 (68.7, 79.9) | 323/356 | 90.9 (87.2, 94.6) |
| Number of children | | | | | | |
| No children | 235/421 | 61.7 (55.8, 67.7) | 85/337 | 23.9 (19.1, 28.9) | 111/356 | 30.9 (25.6, 36.5) |
| One | 90/421 | 18.1 (13.7, 22.5) | 112/337 | 30.3 (24.8, 35.9) | 120/356 | 35.1 (29.8, 40.4) |
| Two | 65/421 | 13.2 (9.8, 16.7) | 84/337 | 25.5 (20.6, 30.3) | 104/356 | 27.9 (23.3, 32.5) |
| Three or more | 31/421 | 6.9 (3.5, 10.4) | 56/337 | 20.3 (15.0, 25.5) | 21/356 | 6.0 (3.5, 8.6) |
| Visited an ANC clinic during most recent pregnancy | 104/263 | 41.6 (32.3,50.9) | 113/254 | 40.4 (33.2, 47.2) | 138/322 | 40.1 (34.6–45.6) |
| Offered an HIV test at the ANC or maternity during most recent pregnancy | 60/104 | 63.3 (46.6, 80.3) | 54/113 | 46.5 (33.4, 57.9) | 21/137 | 13.5 (7.0–19.7) |

[†]Weighted population estimates and 95% confidence intervals (CI).

ANC = Ante-natal care.

during sex with clients in the past 30 day less frequently–by 22.9% of FSW in Colombo, 26.6% in Kandy and 68.4% in Galle.

Around 80% of FSW used condom at last sex with a regular partner in Colombo, 36.5% in Galle, and 16.9% in Kandy, which is substantially less than in 2015. With regular partners every time condom use in the past 30 days was reported by 22.3% in Colombo, 11.5% in Galle and 4.4% in Kandy.

None of the participants in Galle reported ever injecting drugs for non-medical purposes, while this was the case with 4.8% of FSW in Colombo and 0.9% in Kandy.

Ever been sexually assaulted or raped was reported by 10.9% of FSW in Colombo, 1.2% in Galle and 15.5% in Kandy.

## Coverage with HIV prevention services and HIV testing

In 2018, approximately a half of FSW in Colombo and Galle and 39.8% in Kandy reported ever being tested for HIV, which is a slight increase compared to 2015 in Galle and Kandy, and a decline in Colombo (Table 2). In 2018, the commonest reasons in all three cities for never testing for HIV was not knowing where to test (54.2% in Colombo, 41.8% in Galle, 48.1% in Kandy) followed by inconvenient testing location (23.7% in Colombo and 31.1% in Kandy).

The vast majority of FSW in Colombo (80.1%) and Galle (92.6%) were very satisfied or satisfied with the quality of services provided during last HIV testing was received, and 64.2% of FSW in Kandy.

**Table 2. Knowledge about HIV, sexual behaviors and health care utilization among female sex workers in Colombo, Galle and Kandy, Sri Lanka, in 2015 and 2018.**

| | Colombo | | | | Galle | | | | Kandy | | | |
| | 2015 | | 2018 | | 2015 | | 2018 | | 2015 | | 2018 | |
| | n/N | % (95% CI)[†] | n/N | % (95% CI)[†] | n/N | % (95% CI)[†] | n/N | % (95% CI)[†] | n/N | % (95% CI)[†] | n/N | % (95% CI)[†] |
|---|---|---|---|---|---|---|---|---|---|---|---|---|
| **HIV-related knowledge** | | | | | | | | | | | | |
| Ever heard of HIV/AIDS | 474/605 | 77.0 (72.6, 81.1) | 314/454 | 67.2 (61.7, 72.7)* | 235/302 | 78.9 (74.4, 83.7) | 227/354 | 67.3 (61.9, 72.7) * | 300/354 | 85.4 (81.3, 89.7) | 257/362 | 67.6 (62.3, 72.9) * |
| Comprehensive knowledge about HIV prevention[‡] | 195/604 | 31.3 (27.3, 35.3) | 80/454 | 15.0 (10.8, 19.2) * | 84/301 | 28.0 (22.7, 33.3) | 129/354 | 41.5 (35.9, 47.1)* | 183/353 | 50.1 (45.1, 54.8) | 104/353 | 27.8 (22.5, 33.1) * |
| Knows HIV can be transmitted from mother to her unborn child | 376/605 | 59.0 (54.5, 62.3) | 272/357 | 57.5 (51.5, 63.4)* | 209/302 | 68.2 (62.5, 73.8) | 257/360 | 74.5 (69.6, 79.3) | 302/354 | 89.1 (86.6, 92.3) | 259/362 | 68.9 (63.9, 73.8) * |
| Ever heard of ART | 176/605 | 29.6 (25.8, 33.4) | 236/458 | 51.8 (45.4, 58.2)* | 99/300 | 33.0 (26.6, 39.4) | 70/360 | 22.8 (17.3, 28.4)* | 216/354 | 64.2 (59.0, 69.9) | 98/362 | 24.7 (20.1, 29.3)* |
| Knows where to receive an HIV test | 392/605 | 64.1 (60.0, 68.0) | 324/458 | 65.9 (60.0, 71.8) * | 199/302 | 67.0 (60.3, 74.0) | 242/359 | 68.4 (62.6, 74.2) | 229/354 | 63.7 (57.6, 69.5) | 236/357 | 61.2 (55.6, 67.0) |
| **HIV-related behaviours** | | | | | | | | | | | | |
| Perceives personal HIV risk as high | 209/604 | 33.2 (29.1, 37.3) | 111/457 | 19.8 (15.5, 24.1) * | 62/301 | 19.3 (14.2, 24.0) | 69/360 | 14.2 (11.1, 17.2) | 65/354 | 15.5 (10.9, 19.6) | 73/361 | 17.7 (13.8, 21.7) |
| Not discussed HIV with any sexual partner | 262/474 | 57.7 (52.9, 62.7) | 137/314 | 41.6 (34.3, 48.9) | 62/234 | 25.5 (19.6, 31.3) | 164/227 | 65.4 (54.6, 75.7) * | 32/300 | 9.0 (5.3, 12.3) | 129/257 | 52.1 (45.5, 58.8) * |
| Considers male condoms affordable | 511/598 | 85.7 (82.5–89.0) | 274/449 | 65.9 (60.4, 71.4)* | 195/290 | 65.0 (58.1, 71.4) | 190/335 | 61.9 (56.2, 67.6)* | 325/347 | 94.5 (92.2, 97.0) | 58/308 | 19.8 (14.6, 25.0)* |
| Used condom at last sex with a client | 567/604 | 94.0 (92.0, 96.0) | 424/457 | 92.2 (88.9, 95.5) | 269/302 | 87.7 (83.6, 91.6) | 307/360 | 86.6 (83.4, 89.9) | 338/354 | 94.3 (91.2, 97.2) | 207/343 | 57.1 (50.8, 63.2)* |
| Used condom at last sex with a regular partner | 337/375 | 90.4 (87.5, 93.4) | 220/286 | 78.6 (72.6, 84.7)* | 133/151 | 86.4 (80.9, 91.5) | 62/155 | 36.5 (29.0, 44.1)* | 71/152 | 32.3 (22.8, 38.8) | 27/144 | 16.9 (9.8, 23.8)* |
| Ever tested for HIV | 332/605 | 53.0 (48.6, 57.2) | 252/452 | 50.8 (44.8–56.8) | 127/302 | 40.1 (34.0, 45.8) | 175/352 | 49.7 (43.9, 55.6) | 127/354 | 35.6 (29.7, 41.4) | 155/361 | 39.8 (34.4, 45.4) |
| **Health care utilization** | | | | | | | | | | | | |
| Tested for HIV in the past 12 months or knowing HIV positive status | 261/605 | 41.6 (37.2, 45.9) | 158/452 | 31.5 (25.7, 37.3)* | 72/302 | 22.1 (17.1, 26.7) | 127/352 | 39.5 (33.1, 46.0)* | 95/354 | 26.5 (21.2, 31.7) | 70/361 | 17.5 (13.5, 21.5)* |
| Refused health care on the basis of being a FSW | 16/604 | 2.7 (1.2, 4.2) | 75/451 | 16.6 (12.9, 20.4)* | 14/300 | 4.6 (2.3, 6.9) | 12/358 | 2.8 (1.2, 4.4) | 11/354 | 2.3 (0.8, 3.6) | 5/360 | 1.2 (0.2, 2.1) |

[†]Weighted population estimates and 95% confidence intervals (CI)

[‡]All answers correct on five questions related to identifying ways of preventing the sexual transmission of HIV and rejecting major misconceptions about HIV transmission.

ART = anti-retroviral treatment, ANC = antenatal clinic.

* Significant difference ($p<0.05$) in indicator values between the survey carried out in 2018 and the 2015 survey.

Around 40% of respondents visited an ANC during most recent pregnancy but there was a wide variability in reporting being offered an HIV test at an ANC setting–from 13.5% in Kandy to 63.3% in Colombo.

Estimates of coverage with HIV prevention interventions in the past three months were low: 12.5% (95% CI 8.7–16.3) among FSW in Colombo, 15.4% (95% CI 9.8–20.7) in Galle and 9.6% (95% CI 6.5, 12.6) in Kandy. Affordability of male condoms declined when 2018 data are compared to 2015 and was in 2018 reportedly the highest in Colombo (65.9% of FSW reported condoms to be affordable) and the lowest in Kandy (19.8%). The two commonest sources of condoms in Colombo and Galle were private pharmacy and chemist, while in Kandy neighbourhood markets/stands. In Colombo an estimated 21.4% of FSW usually obtain condoms from NGOs and outreach services, and in Galle and Kandy only around 16%.

## Prevalence of HIV, syphilis and HBV

No HIV infections were found in Kandy and Galle while HIV prevalence was 0.4% (95% CI 0.0–1.0) among FSW in Colombo. In Colombo, 8.4% (95% CI 6.3–10.6) of FSW were ever infected with syphilis, as confirmed by a positive TPPA test, while positive VDRL was found among 0.4% (95% CI 0.0–0.9). In Galle, 2.0% (95% CI 0.0, 4.6) of FSW were ever infected with syphilis, and VDRL-reactive were 0.7% (95% CI 0.0–1.6). This was similar to findings in Kandy, with 2.5% (95% CI 0.7–4.2) ever infected with syphilis and 0.6% (95% CI 0.0–1.5) VDRL-reactive. The prevalence of HBsAg positivity was 0.6% (95% CI 0.0–1.3) in Colombo, while no cases were found in the other two cities.

## Correlates of not being tested for HIV in the past 12 months or not knowing HIV positive status

The results of the multivariable analysis show that FSW who did not answer correctly all five questions about HIV transmission had higher odds of not being tested for HIV in the past 12 months or not knowing HIV positive status compared to those who did, with adjusted OR (aOR) of 3.2 (95% CI 1.9, 5.3) in Colombo, 5.1 (95% CI 2.1, 12.4) in Galle and 3.0 (95% CI 1.1, 9.0) in Kandy (Table 3). Divorced, widowed or separated FSW and those who were single had higher odds of not testing compared to married FSW in Colombo. In Galle, FSW with 11–25 paying partners in the last 30 days had marginally higher odds of not being tested compared to FSW with 0–10 partners. In Kandy, FSW who did not use condom at last sex with a non-paying partner had 8.5 (95% CI 3.1, 23.1) higher odds of not being tested, compared to FSW who did. Not using condoms at last sex with a non-paying partner was marginally associated with not being tested for HIV in the past 12 months or not knowing HIV positive status in Colombo and Galle.

## Discussion

We found no HIV infections in Galle and Kandy, and low HIV prevalence in Colombo (0.4%). FSW in Colombo had higher TPPA positivity (8.4%) compared to other two cities. Such findings are encouraging and suggest much lower HIV prevalence among FSW in Sri Lanka compared to the other countries in South East Asia, such as Indonesia (10% in direct and 3% in indirect FSW), India (2.2% at the national level, ranging from 0.7–7.4% across regions) and Myanmar (11–25% across cities) [21–23]. This low HIV prevalence among FSW in three cities in Sri Lanka might be explained by potentially still low HIV transmission among the heterosexual population. The HIV case reporting system in Sri Lanka indicates a total of 350 HIV cases that were newly reported during 2018, which is an increase of 23% from the number reported during 2017 [4]. This increase in reported cases is driven by men: in the period 2011–2018 the number of reported cases increased from 78 to 285 in men, respectively, while around 60 cases were annually reported in women.

**Table 3. Factors associated with not being tested for HIV in the past 12 months or not knowing HIV positive status among FSW in Colombo, Galle and Kandy, Sri Lanka: Bivariable and multivariable logistic regression with unadjusted odds ratios (OR) and adjusted ORs (aOR) with confidence intervals (CI), 2018.**

| | Colombo | | | Galle | | | Kandy | | |
|---|---|---|---|---|---|---|---|---|---|
| | Not tested in the past 12 months or not knowing HIV positive status/ No total | OR (CI 95%) | aOR[a] (CI 95%) | Not tested in the past 12 months or not knowing HIV positive status /No total | OR (CI 95%) | aOR (CI 95%) | Not tested in the past 12 months or not knowing HIV positive status /No total | OR (CI 95%) | aOR (CI 95%) |
| Age | | p = 0.088 | | | p = 0.002 | 1.0 (0.9–1.1)[†] | | p = 0.683 | 0.9 (0.9–1.0)[†] |
| 18–24 | 23/35 | 1 | 1 | 18/19 | 1 | | 20/27 | 1.0 | |
| 25–40 | 140/199 | 1.2 (0.6, 2.7) | 1.4 (0.6, 3.7) | 125/192 | 0.1 (0.01, 0.8) | | 118/146 | 1.4 (0.6, 3.8) | |
| >41 | 131/218 | 0.8 (0.4,1.7) | 0.9 (0.3–2.2) | 82/141 | 0.08 (0.01, 0.6) | | 153/188 | 1.5 (0.6, 3.9) | |
| Monthly income | | p = 0.770 | | | p = 0.143 | | | p = 0.015 | |
| <100 USD | 217/336 | 1 | - | 153/228 | 1 | – | 158/207 | 1.0 | 1.0 |
| >101 USD | 76/115 | 1.1 (0.7,1.7) | | 71/120 | 0.7 (0.4, 1.1) | | 133/154 | 1.9 (1.1, 3.4) | 1.1 (0.4–3.4) |
| Marital status | | p = 0.004 | | | p = 0.519 | | | p = 0.031 | |
| Married | 114/201 | 1 | 1 | 157/252 | 1 | - | 134/175 | 1.0 | 1.0 |
| Single, never married | 40/56 | 1.9 (1.0, 3.6) | 2.2 (1.0, 4.7)* | 21/30 | 1.4 (0.6, 3.2) | | 24/33 | 0.8 (0.4, 1.9) | 1.5 (0.3, 7.9) |
| Divorced, widowed, separated | 140/195 | 1.9 (1.3, 3.0) | 2.2 (1.2, 3.8) | 47/69 | 1.3 (0.7, 2.3) | | 131/151 | 2.0 (1.1, 3.6) | 2.0 (0.5, 8.0) |
| Number of partners in the past 30 days | | p = 0.002 | | | p = 0.054 | | | p = 0.242 | - |
| 0–10 | 34/69 | 1 | 1 | 67/120 | 1 | 1 | 112/132 | 1.0 | |
| 11–25 | 154/239 | 1.9 (1.1, 3.2) | 1.1 (0.5, 2.4) | 101/144 | 1.9 (1.1, 3.1) | 3.3 (1.1, 10.3)* | 172/219 | 0.6 (0.4, 1.2) | |
| >26 | 106/144 | 2.9 (1.6, 5.2) | 1.0 (0.4, 2.5) | 57/88 | 1.5 (0.8, 2.6) | 1.1 (0.4, 3.3) | 7/10 | 0.4 (0.1, 1.7) | |
| Condom use at last sex with a non-paying partner | | p = 0.0175 | | | p = 0.025 | | | p<0.001 | 1.0 |
| Yes | 150/244 | 1 | 1 | 34/60 | 1 | 1 | 18/33 | 1.0 | 8.5 (3.1, 23.1)* |
| No | 50/65 | 2.1 (1.1, 3.9) | 1.9 (1.0, 3.8)* | 71/96 | 2.1 (1.1, 4.3) | 2.1 (1.0, 4.6)* | 114/124 | 9.5 (3.7, 24.4) | |
| Places where clients are sought | | p = 0.0181 | | | p<0.001 | | | p = 0.144 | |
| | | | | | | | | | |
| Fixed locations[‡] | 175/257 | 1 | - | 112/163 | 1 | 1 | 62/76 | 1.0 | - |
| Outdoor places | 81/144 | 0.6 (0.4, 0.9) | | 62/83 | 1.3 (0.7, 2.5) | 1.0 (0.4, 2.5) | 100/137 | 0.6 (0.3, 1.2) | |
| Internet, phone[§] | 36/48 | 1.4 (0.7, 2.8) | | 51/106 | 0.4 (0.3, 0.7) | 0.9 (0.3, 2.5) | 128/145 | 1.7 (0.8, 2.7) | |
| Knowledge about HIV (GAM)[¶] | | p<0.0181 | | | p<0.001 | | | p<0.001 | |
| Score 5 | 33/79 | 1 | 1 | 54/123 | 1 | 1.0 | 53/ 104 | 1.0 | 1.0 |
| Scores 0–4 | 257/369 | 3.2 (1.9, 5.3) | 3.9 (1.9, 8.0)* | 165/223 | 3.6 (2.3, 5.8) | 5.1 (2.1, 12.4)* | 229/ 248 | 11.6 (6.3, 21.3) | 3.0 (1.1, 9.0)* |

[†]Age was included in the multivariate model as a continuous variable due to p-values getting close to the value of 1 if age was treated as a categorical variable.

- not significant in the bivariable analysis and therefore not included in the multivariable logistic regression model.

[‡] Fixed locations: brothel, bar, hotel, motel, spa, massage parlour; Outdoor places: street, park, truck stops;

[§]Also includes finding partners through an intermediary (pimp) and through friends;

[¶] Score 5 implies that answers about all five questions about HIV transmission were correct;

* significantly associated with the outcome in the adjusted multivariate regression model

Our results suggest that the proportion of FSW reporting ever been tested for HIV somewhat increased from 2015 to 2018 in Galle and Kandy, and declined slightly in Colombo. HIV testing in the past 12 months or knowing HIV positive status remains low with only 17.5%-39.5% of FSW across cities reporting testing for HIV in the last 12 months or knowing HIV positive status. Although HIV prevention services for key populations were scaled up in the period 2015–2018, IBBS findings imply that certain sexual risk practices were more prevalent in 2018 compared to 2015. Use of condoms at last sex with clients in Kandy and at last sex with regular partners in all three cities declined when 2015 is compared to 2018. As reported by survey respondents, affordability of condoms significantly decreased from 2015–2018 in all three cities, which could be an explanation for lower condom use. In addition, comprehensive knowledge about HIV prevention significantly declined when the 2015 IBBS findings are compared to 2018. This points to a need for more effective delivery of HIV prevention and testing services to FSW and putting emphasis on achieving much better coverage with HIV prevention. However, some of the differences in reported sexual behaviours, HIV knowledge and HIV testing between the 2015 and 2018 IBBS could be explained by differences in socio-demographic characteristics of FSW recruited in these two rounds of surveys. This might be in particular relevant for Kandy as in the 2018 survey respondents were somewhat older and of lower income.

Multivariate analysis highlighted the importance that adequate knowledge of HIV has on higher uptake of HIV testing, which is consistent with the findings of prior international research [24–26]. The results demonstrate that FSW who did not use condoms at last sex with a non-paying partner had lower odds of testing for HIV. These findings complement those from other studies, which found a negative association between inconsistent condom use with non-paying partners and HIV testing, possibly due to lower risk perception of women who are in longer-term relationships [27–30]. Condom use with regular partners was substantially lower compared to paying partners, as has repeatedly been shown elsewhere, despite the fact that most FSW did not know the HIV status of their partners [31, 32]. Respondents in our study were reporting considerably more frequently condom use at last sex with paying and non-paying partners compared to consistent condom use during a 30 days time-frame. As suggested by Baral et al., always using condoms may be a more representative measure of actual condom use compared to condom use at last sex [33].

In Colombo, FSW who were married were more likely to get tested, which might be explained by their higher motivation to stay healthy and support their families [34, 35].

The systematic review of barriers and facilitators to HIV testing amongst FSWs found that the two barriers most commonly reported were costs (including transportation, formal/informal payments) and stigma [36]. In our study, the most important reason given for not testing for HIV was not knowing where to test. Therefore, raising awareness of HIV testing via targeted campaigns along with educational activities should be the cornerstone of HIV prevention efforts among FSW in Sri Lanka. Innovative testing approaches are needed to reach FSW with HIV testing such as mobile testing services, social-network testing, self–testing and, importantly, testing of partners [8]. Interventions for FSWs should promote consistent condom use, increase skills in negotiating condom use across all types of partnerships and self-efficacy of sex workers when interacting with clients and regular partners. It is also important to reach with HIV prevention more hidden FSW that are not finding partners at fixed locations and at public places. The percentage of FSW who reported finding partners through mobile phone, Internet or an intermediary ranged from 10.9% in Colombo to 38.7% in Kandy.

We also found geographical disparities in utilization of HIV prevention among FSW, with the poorest utilization in Kandy. Sub-optimal coverage with HIV interventions is likely due to both insufficient levels of service provision but also limited uptake of services because of

clandestine nature of sex work and stigma. Other reasons could be underlying socio-economic differences among these geographical areas in Sri Lanka. As found in the latest DHS, proportion of women of reproductive age who are illiterate is higher in Kandy compared to Colombo and Galle, and a lower proportion are employed. Findings from our study also indicate lower earnings of FSW in Kandy compared to two other cities, which might have an impact on utilization of HIV prevention services.

The majority of women in this study had an occupation outside of sex work, which might suggest that they engage in occasional transactional sex based on financial need. Addressing financial and social vulnerabilities of these women is therefore of utmost importance. The results highlight the need to address the quantity of service provision and focus on empowerment–based approaches that address the socio-structural vulnerabilities of FSWs to HIV [37].

## Study limitations

There are a number of limitations to our study. FSW provided self-reported data, which could be subject to recall bias. Social desirability bias could impact on results, particularly on reporting of sexual behaviours. This source of bias might have been limited due to the engagement of trained study personnel who were experienced in working with FSWs. Comparisons of surveys carried out in 2015 and 2018 is impacted by the extent of those biases, as well as selection bias and different socio-demographic characteristics of respondents.

The use of RDS increases the generalisability of the results by adjusting for network size, homophily, and by limiting recruitment per respondent.

## Conclusions

Findings from this study reveal the similarities and heterogeneities in vulnerabilities towards HIV among FSWs in the three cities in Sri Lanka. HIV has not yet been firmly established among FSW in these cities, which provides a window of opportunity to avoid progression of HIV transmission. This can be achieved by larger-scale implementation of behavioural, biomedical and structural interventions that are known to be effective.

## Supporting information

**S1 File.**
(SAV)

**S2 File.**
(SAV)

**S3 File.**
(SAV)

## Acknowledgments

We would like to thank the study participants and the staff of the National STD/AIDS Control Programme in Sri Lanka, Management Frontiers Ltd., and the staff from the following NGOs that participated in data collection: Abhimani, Saviya Development, Laksetha Sahana Sewa, Family Planning Association, Wayamba Govi Sanwardhana Padanama, Rajarata Gami Pahana, Saviya Development Foundation, Natural Resource Development Foundation and Sri Lanka Human Development Foundation.

## Author Contributions

**Conceptualization:** Ariyaratne Manathunge, Ivana Bozicevic.

**Formal analysis:** Jelena Barbaric, Tomislav Mestrovic, Ivana Bozicevic.

**Funding acquisition:** Ariyaratne Manathunge, Sriyakanthi Beneragama.

**Methodology:** Ivana Bozicevic.

**Project administration:** Sriyakanthi Beneragama.

**Resources:** Ariyaratne Manathunge, Sriyakanthi Beneragama.

**Supervision:** Ariyaratne Manathunge.

**Writing – original draft:** Ariyaratne Manathunge, Jelena Barbaric, Tomislav Mestrovic, Sriyakanthi Beneragama, Ivana Bozicevic.

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
