## [Decision Letter · Decision Letter 0]

14 Jul 2020

PONE-D-20-14017

HIV prevalence, sexual risk behaviours and HIV testing among female sex workers in three cities in Sri Lanka: Findings from respondent-driven sampling surveys

PLOS ONE

Dear Dr. Bozicevic,

Thank you for submitting your manuscript to PLOS ONE. After careful consideration, we feel that it has merit but does not fully meet PLOS ONE’s publication criteria as it currently stands. Therefore, we invite you to submit a revised version of the manuscript that addresses the points raised during the review process.

We look forward to receiving your revised manuscript.

Kind regards,

Zixin Wang, PhD.

Academic Editor

PLOS ONE

Journal Requirements:

2. Please amend your current ethics statement to address the following concerns: Please explain why written consent was not obtained, how you recorded/documented participant consent, and if the ethics committees/IRBs approved this consent procedure.

3. Please provide more information on the coupons given out as part of recruitment.

Reviewers' comments:

Reviewer's Responses to Questions

**Comments to the Author**

1. Is the manuscript technically sound, and do the data support the conclusions?

Reviewer #1: Yes

Reviewer #2: Yes

2. Has the statistical analysis been performed appropriately and rigorously? 

Reviewer #1: Yes

Reviewer #2: No

3. Have the authors made all data underlying the findings in their manuscript fully available?

Reviewer #1: Yes

Reviewer #2: Yes

4. Is the manuscript presented in an intelligible fashion and written in standard English?

Reviewer #1: Yes

Reviewer #2: No

5. Review Comments to the Author

Reviewer #1: The authors conducted a second round of IBBS survey among female sex workers in Colombo, Galle, and Kandy, and provided valuable results about HIV prevalence, risk behaviors, and related perceptions. This study adds to our knowledge about the HIV/AIDS epidemic in Sri Lanka, therefore of significance. I would agree for the manuscript to be published if the authors can provide more detail for the following suggestions.

Major:

Could the author explain the reason for choosing the three cities as the research site? How representative are the results of these three cities?

Minor:

1.There is a lack of background data on sexual risk behaviors and HIV testing among female sex workers in Sri Lanka in the introduction part. Please provide more data if possible.

2.Please clarify the reason for choosing “consistent condom use with a non-paying partner for 30 days” as the indicator in sample size estimation.

3.Please add more details of the measurement of Sexual risk behaviors since it is one of the major outcomes.

4.Please clarify why only the result of Colombo was weighted in table 1.

5.Please provide the statistical testing results (like P-value) in the comparison between two rounds of the survey in different cities in table 2. (2015 versus 2018)

6.Please mark the significance of aOR in table 3.

Overall, the manuscript is well written and showed us a more up-to-date picture of HIV infection and perception among female sex workers in Sri Lanka. Apart from the above problems, language and format should be improved before getting published.

Reviewer #2: Their investigations conducted a study in Sri Lanka, and described the HIV, syphilis and hepatitis B (HBV) prevalence, sexual risk behaviors and utilization of HIV prevention interventions among female sex workers (FSW), this study could provide more information of interventions for FSW in Sri Lanka. I have several comments for the authors to consider.

1. In the introduction, the authors should introduce why they chose the three places (Colombo, Galle and Kandy) as the study sites?

2. The authors should provide a further implication of this study, not only to monitor and further characteristics HIV-related risk behaviors, uptake of HIV prevention services and HIV prevalence?

Results

1. Is the FSW collected in 2015 and 2018 the same group? Is there any difference between the FSW in 2015 and 2018? Could the authors add the data of 2015 and compared with that of 2018 in Table 1 for the demographic characteristics?

2. The unit of age in Table 1 should be added.

3. Can you rearrange the contents of Table 2 logically, which is very difficult to read. Also, can you compare the knowledge about HIV, sexual behaviors and health care utilization between 2015 and 2018?

4. If the author compares the differences between the three regions, more information may be provided.

Discussion

1. The author should add more information about the three study sites of social economics, which could give more information to explain the differences of factors associated not being tested for HIV in the past 12 months or not knowing HIV positive status among FSW.

2. Selection bias may be a limitation in this study.

6. PLOS authors have the option to publish the peer review history of their article (what does this mean?). If published, this will include your full peer review and any attached files.

Reviewer #1: No

Reviewer #2: No

---

## [Author Response · Author response to Decision Letter 0]

20 Jul 2020

Comments of the Editorial Office and Responses of the authors

Please ensure that your manuscript meets PLOS ONE's style requirements, including those for file naming 

Thank you, this has been corrected.

Please amend your current ethics statement to address the following concerns: Please explain why written consent was not obtained, how you recorded/documented participant consent, and if the ethics committees/IRBs approved this consent procedure. 

Written consent was not obtained as it is a standard practice in HIV bio-behavioural surveys in key populations that written consent is not asked for, but an oral consent. Following is added: 

After they gave an oral consent to participate, a survey staff member signed an informed consent form on their behalf. The Ethics Committee approved this consent procedure. 

Please provide more information on the coupons given out as part of recruitment. 

More information is now provided in Methods:

Each coupon had its unique number that enabled to link participants with those who gave them coupons to participate in the study. Coupon numbers enable to monitor recruitment chains and issue secondary incentives, and are also necessary for RDS data analysis. Coupons also contained information about the address of the study site and time-period when the study site is open. 

 We note that you have indicated that data from this study are available upon request. PLOS only allows data to be available upon request if there are legal or ethical restrictions on sharing data publicly.

 There are no legal or ethical restrictions on sharing these data publicly.

We uploaded the anonymized data set necessary to replicate study finding as part of submission.

Reviewer 1

Major:

Could the author explain the reason for choosing the three cities as the research site? How representative are the results of these three cities?

These three cities were selected for IBBS in 2018 because IBBS were done in 2015 in these three cities. By repeating surveys in the same cities over time, changes in HIV indicators over time can be assessed. These three cities were selected in 2015 (and 2018) because they are in the most populous provinces in Sri Lanka named Western Province (Colombo), Central Province (Kandy), and Southern Province (Galle) and because of estimated population size of FSW. 

This explanation is added at the end of Introduction.

 Minor:

1.There is a lack of background data on sexual risk behaviors and HIV testing among female sex workers in Sri Lanka in the introduction part. Please provide more data if possible.

Thank you, more data are added on sexual risk behaviours and HIV testing in Introduction. 

2.Please clarify the reason for choosing “consistent condom use with a non-paying partner for 30 days” as the indicator in sample size estimation.

We chose this indicator because this is an important indicator of sexual behaviours in FSW. We could not base sample size calculation on prevalence of infections because of low prevalence estimates from IBBS in FSW in 2015.

3.Please add more details of the measurement of Sexual risk behaviors since it is one of the major outcomes.

This is added now to Methods

4.Please clarify why only the result of Colombo was weighted in table 1.

All prevalence estimates were weighted. This is corrected now in Table 1. 

5.Please provide the statistical testing results (like P-value) in the comparison between two rounds of the survey in different cities in table 2. (2015 versus 2018)

We compared now all indicators collected in the 2015 and 2018 surveys using chi-square test, and denoted with a symbol * in Table 2 where there is significant differences (p<0.05) in values.

6.Please mark the significance of aOR in table 3.

This is now marked with an *

Reviewer 2

1. In the introduction, the authors should introduce why they chose the three places (Colombo, Galle and Kandy) as the study sites?

This is now added to the Introduction.

2. The authors should provide a further implication of this study, not only to monitor and further characteristics HIV-related risk behaviors, uptake of HIV prevention services and HIV prevalence? 

We provided several implications for programmatic interventions, in particular need to expand community-based HIV testing, address financial and social vulnerabilities and focus on empowerment-based approaches. 

We added now a need for interventions to focus on promoting consistent condom use across all types of partnerships and increase skills in negotiating condom use.

 1. Is the FSW collected in 2015 and 2018 the same group? Is there any difference between the FSW in 2015 and 2018? Could the authors add the data of 2015 and compared with that of 2018 in Table 1 for the demographic characteristics?

We added some demographic data on FSW recruited in 2015 to the paragraph in Results under socio-demographic characteristics. We thought that this is better than adding data to Table 1. 

2. The unit of age in Table 1 should be added.

This is added to Table 1

3. Can you rearrange the contents of Table 2 logically, which is very difficult to read. Also, can you compare the knowledge about HIV, sexual behaviors and health care utilization between 2015 and 2018?

We re-arranged the contents of Table 2 so that it is more logical. We compared now all indicators collected in the 2015 and 2018 survey using chi-square test, and denoted with a sign of * in the Table 2 where there is significant differences (p<0.05)

4. If the author compares the differences between the three regions, more information may be provided.

We provided more information on socio-economic differences between Colombo, Galle and Kandy in Introduction. This information was collected in DHS in 2016.

Discussion 

The author should add more information about the three study sites of social economics, which could give more information to explain the differences of factors associated not being tested for HIV in the past 12 months or not knowing HIV positive status among FSW.

As stated above, we provided more information on socio-economic differences between Colombo, Galle and Kandy in Introduction. This information was collected in DHS in 2016.

Implications of these data is discussed in Discussion.

2. Selection bias may be a limitation in this study.

Agree, this is added to Limitations.

---

## [Decision Letter · Decision Letter 1]

25 Aug 2020

PONE-D-20-14017R1

HIV prevalence, sexual risk behaviours and HIV testing among female sex workers in three cities in Sri Lanka: Findings from respondent-driven sampling surveys

PLOS ONE

Dear Dr. Bozicevic,

Thank you for submitting your manuscript to PLOS ONE. After careful consideration, we feel that it has merit but does not fully meet PLOS ONE’s publication criteria as it currently stands. Therefore, we invite you to submit a revised version of the manuscript that addresses the points raised during the review process.

We look forward to receiving your revised manuscript.

Kind regards,

Zixin Wang, PhD.

Academic Editor

PLOS ONE

Reviewers' comments:

Reviewer's Responses to Questions

**Comments to the Author**

1. If the authors have adequately addressed your comments raised in a previous round of review and you feel that this manuscript is now acceptable for publication, you may indicate that here to bypass the “Comments to the Author” section, enter your conflict of interest statement in the “Confidential to Editor” section, and submit your "Accept" recommendation.

Reviewer #1: All comments have been addressed

Reviewer #2: All comments have been addressed

2. Is the manuscript technically sound, and do the data support the conclusions?

Reviewer #1: Yes

Reviewer #2: Yes

3. Has the statistical analysis been performed appropriately and rigorously? 

Reviewer #1: Yes

Reviewer #2: Yes

4. Have the authors made all data underlying the findings in their manuscript fully available?

Reviewer #1: Yes

Reviewer #2: Yes

5. Is the manuscript presented in an intelligible fashion and written in standard English?

Reviewer #1: No

Reviewer #2: No

6. Review Comments to the Author

Reviewer #1: The authors have addressed all of the questions from the last review.

But it would be better if the author can provide more information regarding the following suggestion:

1. Sample size calculation:

the author explained in the last response that using ‘condom use with a non-paying partner’ to calculate sample size was due to the importance of condom use, however, there are several behavioral indicators including testing for HIV/knowing one’s status, which is the outcome of the regression. It would be better if the author could calculate sample sizes with other indicators and compare them with the current sample size to demonstrate the power of the current choice. If other calculated sample sizes were larger, limitations should be addressed in discussion.

2. The outcome:

First, the outcome of regression is “not testing or not knowing one’s status”, yet in table 2, there was one variable named “tested or know”. It would be good if the author can clarify if the rates of these two variables are complementary and the sum of them is 1. If so, should the outcome of regression be “not testing and not knowing”? Please ensure the accuracy of the expression.

Second, as was mentioned in the discussion, the rate of “Recent HIV testing” ranged from 17.2% to 39.5%. However, after reading table 2 carefully, the variables that match this expression is “tested for HIV in the past 12 months or knowing HIV positive status”. Though the HIV positive rates were low in this study, it was not appropriate to mix these two indicators in the abstract and discussion. Please revise.

Third, the number of 17.2% in the second suggestion wasn’t displayed in table2. The closest number was 17.1 and 17.5. Please make sure there is no typo.

3. The discussion:

The author provided statistical testing between data in 2015 and 2018. And there were many significant results, which haven’t been referred to or deeply discussed in the discussion part. It would be better if the author could provide more insight into the reason of changes from 2015 to 2018, as well as implication for policy makers.

4. Uses with dots and commas:

In paragraph one of introduction, both line one and line five have displayed a confusing mixture of dots and commas. Please proofread the paper and make sure such uses are consistent within the manuscript.

Reviewer #2: The author has made a good modification according to the comments, but some minor revisions are still should be taken.

Introduction:

In the last paragraph, the author should write the research hypothesis rather than the results of the demographic and health survey.

In Table 2

The first column can be classified by knowledge, behavior and health care utilization.

7. PLOS authors have the option to publish the peer review history of their article (what does this mean?). If published, this will include your full peer review and any attached files.

Reviewer #1: No

Reviewer #2: No

---

## [Author Response · Author response to Decision Letter 1]

10 Sep 2020

We provided Responses to reviewers as a separate file

Comments of Reviewer 1 

 Responses of the authors

1. Sample size calculation:

the author explained in the last response that using ‘condom use with a non-paying partner’ to calculate sample size was due to the importance of condom use, however, there are several behavioral indicators including testing for HIV/knowing one’s status, which is the outcome of the regression. It would be better if the author could calculate sample sizes with other indicators and compare them with the current sample size to demonstrate the power of the current choice. If other calculated sample sizes were larger, limitations should be addressed in discussion.

We did the calculation of sample sizes for several other indicators including “tested for HIV in the past 12 months or knowing one’s status” and “comprehensive knowledge about HIV prevention” but that yielded smaller sample sizes compared to ‘condom use with a non-paying partner’. 

Used condom at last sex with a client was not used to calculate the sample size as the value of this indicator was already high in the 2015 IBBS, so a large sample size would have been needed to detect a small change in this indicator. 

Final samples sizes that we calculated and the number of FSW recruited - 458 FSW in Colombo, 360 in Galle and 362 in Kandy – is comparable to IBBS done in KPs in other settings. 

2. The outcome:

First, the outcome of regression is “not testing or not knowing one’s status”, yet in table 2, there was one variable named “tested or know”. It would be good if the author can clarify if the rates of these two variables are complementary and the sum of them is 1. If so, should the outcome of regression be “not testing and not knowing”? Please ensure the accuracy of the expression.

Second, as was mentioned in the discussion, the rate of “Recent HIV testing” ranged from 17.2% to 39.5%. However, after reading table 2 carefully, the variables that match this expression is “tested for HIV in the past 12 months or knowing HIV positive status”. Though the HIV positive rates were low in this study, it was not appropriate to mix these two indicators in the abstract and discussion. Please revise.

Third, the number of 17.2% in the second suggestion wasn’t displayed in table2. The closest number was 17.1 and 17.5. Please make sure there is no typo. The outcome of regression is “not being tested for HIV in the past 12 months or not knowing HIV positive status”. This is in the title of the Table 3. 

Now, we added this clarification to the the columns of Table 3.

In Table 2, the outcome is “Tested for HIV in the past 12 months or knowing HIV positive status”. 

The sum of these proportions is 1. 

This is right, we corrected this so that instead of “recent HIV testing” it stands “tested for HIV in the past 12 months or knowing HIV positive status”. 

It should be 17.5% - this is now corrected in Abstract and Discussion 

3. The discussion:

The author provided statistical testing between data in 2015 and 2018. And there were many significant results, which haven’t been referred to or deeply discussed in the discussion part. It would be better if the author could provide more insight into the reason of changes from 2015 to 2018, as well as implication for policy makers. 

We provided a discussion about significant differences in some results, in particular HIV testing, condom use and comprehensive knowledge about HIV prevention. 

The implication of results in terms of development of HIV response in FSW is addressed later on in Discussion. 

Uses with dots and commas:

In paragraph one of introduction, both line one and line five have displayed a confusing mixture of dots and commas. Please proofread the paper and make sure such uses are consistent within the manuscript. 

This was corrected in the Introduction, and checked in other parts of the manuscript. 

Reviewer 2 

Introduction:

In the last paragraph, the author should write the research hypothesis rather than the results of the demographic and health survey. 

Research hypothesis was added to the Introduction (We hypothesised that there will be a significant increase in condom use with non-paying partners and ever testing for HIV when 2015 data are compared with 2018.)

During the first round of review, the other reviewer suggested to add results on demographics in these three cities. 

In Table 2

The first column can be classified by knowledge, behavior and health care utilization. 

This is added now to classify indicators

---

## [Editor Report · Decision Letter 2]

16 Sep 2020

HIV prevalence, sexual risk behaviours and HIV testing among female sex workers in three cities in Sri Lanka: Findings from respondent-driven sampling surveys

PONE-D-20-14017R2

Dear Dr. Bozicevic,

We’re pleased to inform you that your manuscript has been judged scientifically suitable for publication and will be formally accepted for publication once it meets all outstanding technical requirements.

Kind regards,

Zixin Wang, PhD.

Academic Editor

PLOS ONE
---

## [Editor Report · Acceptance letter]

21 Sep 2020

PONE-D-20-14017R2 

HIV prevalence, sexual risk behaviours and HIV testing among female sex workers in three cities in Sri Lanka: Findings from respondent-driven sampling surveys 

Dear Dr. Bozicevic:

I'm pleased to inform you that your manuscript has been deemed suitable for publication in PLOS ONE. Congratulations! Your manuscript is now with our production department. 

Kind regards, 

on behalf of

Professor Zixin Wang 

Academic Editor

PLOS ONE